# Do Private Transport Services Complement or Compete against Public Transit? Evidence from the Commuter Vans in Eastern Queens, New York

**Catherine Musili [1] and Deborah Salon [2],\***

[1]    School of Sustainability, Arizona State University, Tempe, AZ 85281, USA; Catherine.Musili@asu.edu
[2]    School of Geographical Sciences and Urban Planning, Arizona State University, Tempe, AZ 85287, USA
\*    Correspondence: Deborah.Salon@asu.edu

**Abstract:** Do private transport services complement or compete against public transit? As transit agencies scramble to adjust to the new transport landscape of mobility services, this has become an important question. This study focuses on New York's commuter vans (also known as "dollar vans"), private vans that have operated alongside public transit for decades. We use original survey and observational data collected in the summer of 2016 to document basic ridership characteristics and to provide insight into whether the commuter vans complement or compete against city buses. Commuter van ridership in Eastern Queens is high; it is roughly equivalent to city bus ridership on parallel routes at approximately 55,000 per day. Further, more than 60% of van riders surveyed would have had a free trip on a city bus, through either a transit pass or transfer. Time savings was an important motivation for these riders to pay extra for the vans; the vans are faster than city buses, and van wait times are shorter. These results suggest that New York's commuter vans complement public transit by serving as a feeder system. This conclusion, however, is highly context-dependent. As private transport services proliferate, continued research is needed to ascertain their relationships with public transit.

**Keywords:** paratransit; dollar vans; jitneys; informal transport; New York City; microtransit; ride hailing

---

## 1. Introduction

New York's commuter vans—also known as "dollar vans" or jitneys—carry approximately 120,000 passengers each day [1]. This is comparable to the total transit ridership in other US cities such as Phoenix, AZ [2] and Minneapolis/St. Paul, MN [3]. Despite its size, little is known about the commuter van industry. Because these vans are privately owned and operated, data that is commonly available for transit systems is absent. This study is one of the first to shed light on New York's "shadow transit" system, building on the work of King and Goldwyn [1], King [4], Reiss and Lavey [5], and Goldwyn [6].

Commuter vans occupy a market space in between taxis/ride hailing services and formal public transit. They are faster than city buses and offer some of the flexibility, comfort, and convenience of neighborhood taxis for an affordable US $2 fare. They have an intermediate capacity of 13–20 passengers and follow a semi-regular route. The vans pick up passengers along their route like city buses, but are often street-hailed like taxis. They can deviate from the route at the driver's discretion, but can only legally serve a predetermined neighborhood area.

With these features, commuter vans can complement public transit by serving as a feeder system for the buses and subways that transport passengers out of the van-served neighborhoods.

However, they can also directly compete against local city buses. Some researchers have argued that while services such as ride hailing and commuter vans threaten public transit, they are also key to its future success [7]. These private services help to smooth transportation demand shocks caused by temporary transit disruptions such as closing of subway stations [8], extreme weather conditions, and unexpected natural disasters as in the case of New York City's Hurricane Sandy [9].

The perception of the role played by services such as commuter vans is at the core of policy decisions surrounding these services. For instance, the New Jersey Interstate Commerce Commission views jitneys as contributing to overall increased mobility even though they are in direct competition with New Jersey transit [10], hence it implemented policies that encouraged new entries into the market and relaxed jitney restrictions [10]. Stricter policies like those that govern New York commuter vans, such as levying fines for operation of commuter vans along Metropolitan Transportation Authority (MTA) bus routes, are the result of the perception that commuter vans compete against the city's MTA transit.

This research documents travel and sociodemographic characteristics of commuter van riders in Eastern Queens and uses these data to shed light on the role that these vans play in relation to the formal MTA transit system in the study area. In particular, we ask four research questions:

1. How many people ride commuter vans in Eastern Queens?
2. What are the socioeconomic characteristics of commuter van riders?
3. Why do riders choose the commuter vans over public buses?
4. Do vans complement or compete against MTA transit service?

The first two questions are motivated by the need to understand both the extent of commuter van services and the population they serve. The third question is motivated by a need to understand how commuter van and MTA transit services compare from the van riders' point of view. Answering the last question is important for making evidence-based commuter van policy going forward. Our answers to these questions are drawn from original on-board surveys and a tally of commuter van riders in Eastern Queens, conducted in the summer of 2016.

This research is especially timely today, as ride hailing service providers such as Uber and Lyft are moving into the collective transport business. For instance, uberPOOL and Lyft Line provide service to up to 4 or 6 passengers at a time, and Uber's Express POOL, Lyft Shuttle, and services such as Via require people to walk to a meeting point comparable to a bus stop and offer fixed routes and fixed pricing. These new jitney-style services are remarkably similar to New York's commuter vans, and questions have been raised about the extent to which they compete against both the more heavily-regulated taxi industry [11,12] and against public transit [13–15]. City approaches to regulating these new services are still evolving, but decision making is hampered by limited information about how these services affect travel patterns [16]. This study of New York's commuter vans—a private jitney-style service that has existed alongside MTA transit for over three decades—can inform these regulatory decisions.

## 2. New York's Commuter Vans

Jitneys operate in almost every city in the developing world, and in many wealthier cities as well. Many jitney services in US cities sprang up in response to inefficient, inconsistent, or simple lack of formal public transit services in the neighborhoods they serve (e.g., Jersey City [17], Pittsburg [18], San Francisco [19], and Miami [20]). In the US, they are often viewed as informal services run mostly by immigrants for immigrants [21], providing low-cost collective transport services to low-income neighborhoods with low car ownership.

The New York commuter vans fit this description. Neighborhood vans in the Bronx, Brooklyn, and Eastern Queens are largely owned and operated by recent immigrants from the Caribbean and Africa, while the vans operating between Chinatowns in Manhattan, Brooklyn, and Queens are owned and operated by Chinese immigrants. The heavy immigrant influence in van ownership and operation,

however, does not limit the service to foreign born riders. Our data show that only half of commuter van riders in Eastern Queens are foreign born. That said, fieldwork observations clearly indicated a racial pattern among van riders; the vast majority were of African or Hispanic descent, both US-born and foreign-born. In fact, in three months of data collection requiring daily interaction with van riders and drivers, we observed only one Caucasian van rider in a neighborhood that is just over 10% white [22]. Although the survey administered in this study did not specifically ask for the racial/ethnic identification of the van riders, about 50% of the survey respondents indicated that they were immigrants from the Caribbean (65%), Africa (25%), Latin America (8%), and Asia (2%).

New York commuter van service began in 1980 in response to an 11-day MTA labor strike. The strike opened an opportunity for people with personal vehicles to step in and provide transport services for a small fee. This opportunity was lucrative enough that even after the MTA strike ended, some continued to offer transport services, albeit illegally. In 1982, the New York State Department of Transportation began to grant permits to commuter vans, making them legal but doing little else to ensure compliance with regulations [6]. In 1993, the New York City Taxi and Limousine Commission (TLC) was granted regulatory authority over commuter vans and remains the governing body to date. New York presents a somewhat unusual case because although the vans sprang up due to inadequate or absent formal transit services in certain neighborhoods, most of these neighborhoods today have MTA transit service comparable in quality to that in other parts of New York.

Commuter vans today remain privately owned and operated. They provide low-cost collective transport service within selected neighborhoods in New York, and they do not receive government subsidies. Their fare is regulated to be $2 per ride. Most commuter vans in the study area operate from 06:00 or 07:00 to 18:00 or 19:00 with some operating earlier or later. By law, these individually-owned commuter vans must be registered under a commuter van authorization or base "company". A commuter van base is a TLC-licensed business that sends TLC-licensed commuter vans to transport passengers within a particular geographic service area in New York City [23]. In some cases, the base license holder also owns all the vans under his/her base and rents them out to drivers. In other cases, van owner-operators pay a fee to join an existing commuter van base [6]. In August 2016, there were 443 licensed commuter vans operating in New York City [24], 270 of which were operating in our study area of Eastern Queens [25].

Like jitney services nearly everywhere, the New York commuter vans present a regulatory challenge for the city. While regulating to ensure public safety, a balance must be struck between encouraging these transport services to complement public transit and discouraging them from stealing transit riders and revenues. Achieving this regulatory balance is especially difficult without detailed van ridership data, such as that presented in this article. Many of the current rules governing commuter vans focus on combating transit rider stealing. Specifically, commuter vans are licensed only to provide prearranged transport (street hail pickups are not allowed), are not allowed to use the same routes as MTA buses, and are not allowed to pick up passengers at bus stops. Because their business model relies on street hails and it is difficult to create viable van routes that do not overlap with bus routes, these rules have pushed the van industry to operate only semi-legally. There are semi-regular TLC "crackdowns" during which van drivers and owners receive costly tickets for failing to follow these rules.

As we will demonstrate in this article, however, the commuter vans actually play a largely complementary role to MTA transit in Eastern Queens. Discouraging the commuter van industry likely hurts both access and mobility in the van-served neighborhoods and adds to already-crowded conditions on MTA buses in the neighborhood.

## 3. Eastern Queens Study Area Characteristics

Eastern Queens commuter vans serve six main routes in both Queens and Nassau County [5]. This study focuses on the five commuter van routes operating in Queens County leaving out the one route operating solely in Nassau County. These commuter vans connect five neighborhoods in Eastern

Queens to the Jamaica Center commercial district: Rosedale, Cambria Heights, Green Acres Mall, Far Rockaway and Linden. Additional neighborhoods are served as commuter vans operate on roads through or adjacent to them.

The neighborhoods served by the commuter vans include 115 census tracts. Figures 1 and 2 illustrate the van routes and the location of the study area in the context of New York City and the MTA transit system, as well as the median household income, population density, and the percent of households that own cars [22]. The study area is the easternmost part of New York City adjacent to Long Island, with a suburban feeling. In comparison with much of New York City, the area is more sparsely populated, and the north and west sides are more densely populated than the rest of the study area. The median household income is higher in the sparsely populated east and south side of the study area, while the higher density west side houses residents with lower income. Still, the median income in most of the study area is actually higher than New York City's median annual income of about $53,000.

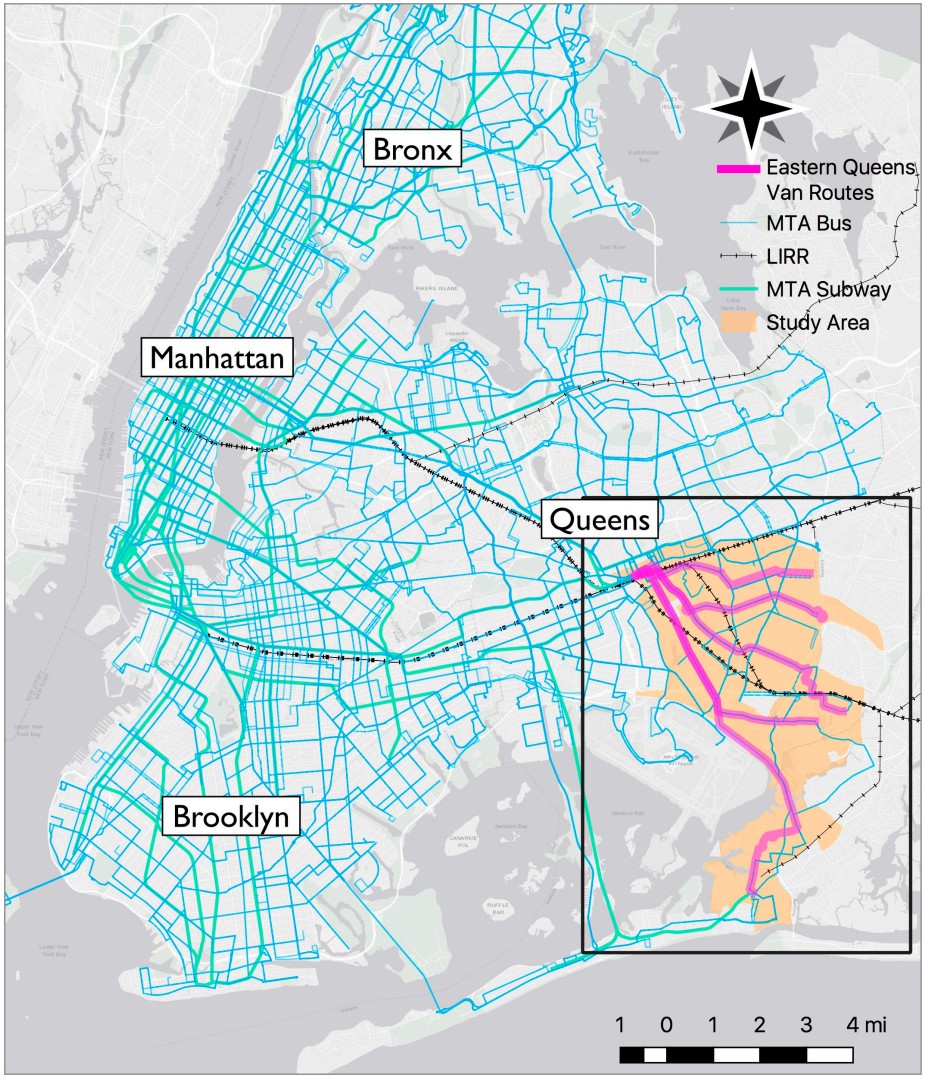

**Figure 1.** The study area, Metropolitan Transportation Authority (MTA), and commuter van routes.

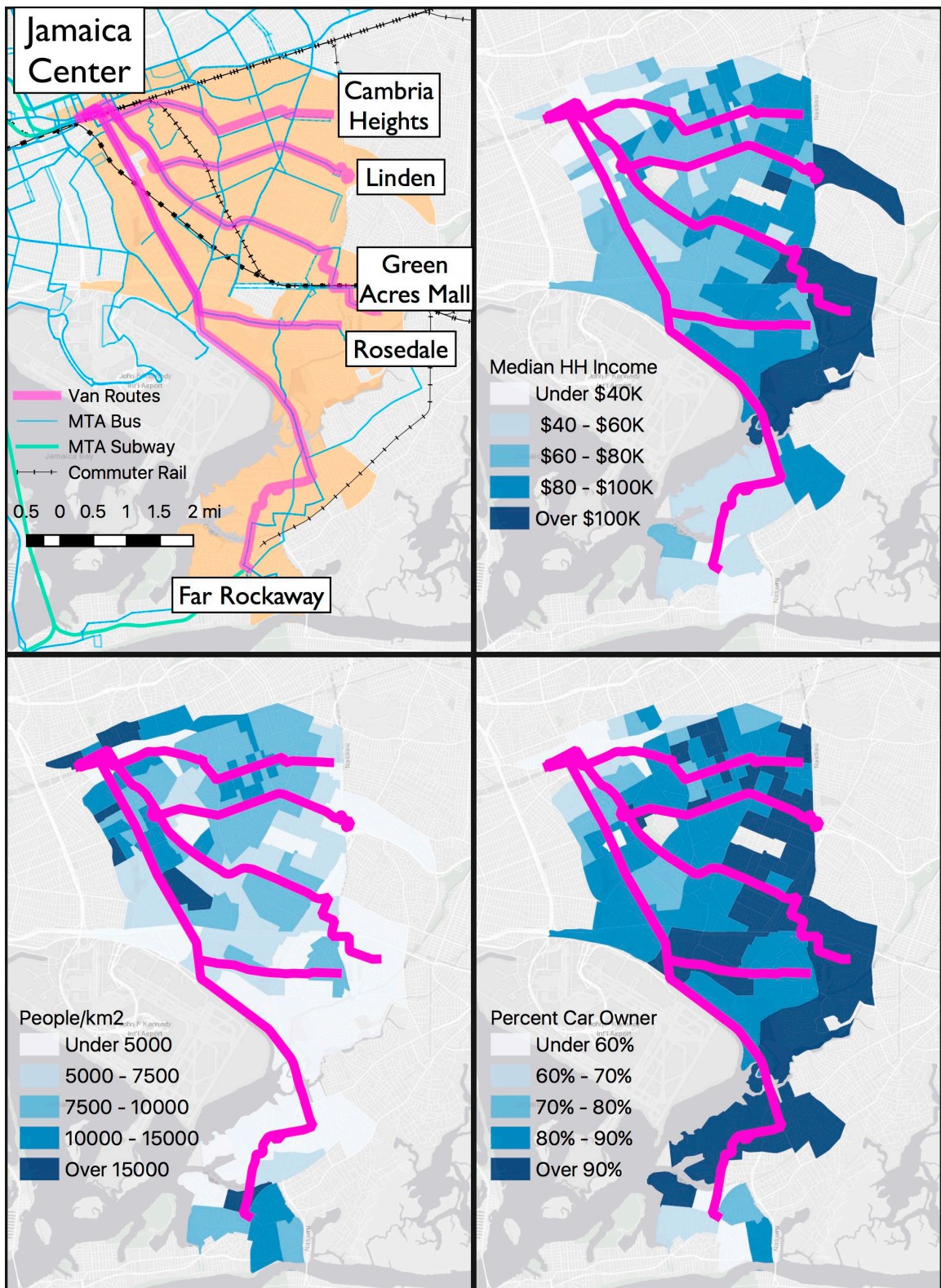

**Figure 2.** Clockwise from top left: commuter van routes and MTA transit, household income, car ownership, and population density [22].

The bottom right map in Figure 2 depicts the spatial distribution of household car ownership in the study area. Households without cars are completely dependent on shared modes of transportation. In the northwest corner of the study area where density is high and incomes are low, less than half of

households own cars. This vehicle ownership pattern directly impacts the spatial pattern of commute mode choices. Where car ownership is low, people are more likely to ride public transportation. In census tracts where car ownership is higher, commute trips are made by carpooling or driving alone. Note, however, that by the census definition, a commuter van trip is a "carpool" trip. For the longest portion of their commute, the census reports that 43% of commuters in the study area used public transportation, 6% carpooled, and 44% drove to work in 2016. As will become clear from our data, however, many commuters use commuter vans *together with* MTA transit to get to work.

## 4. Materials and Methods

Data for this study were collected over three months in summer of 2016 using an Arizona State University Institutional Review Board-approved survey questionnaire (Appendix A provides a copy of the survey and raw data totals for each question), a three-day van rider tally on each of the five Eastern Queens routes, and researcher observation. Surveys were administered on board randomly-selected commuter vans. Survey implementation required the researcher to ask drivers for permission to ride in their vans and talk to their passengers, after paying the $2 fare for the trip. The researcher then gave all passengers the opportunity to participate in the survey and rode each van until all surveys were returned or to the end of the route, whichever came first. The survey response rate was approximately 40%.

In addition to the surveys, a 3-day ridership tally was conducted on each of the five routes in the study area. The methodology used was simple, but time-intensive; the researcher rode one van from approximately 07:00 to 19:00 for three consecutive days, counting each passenger and noting their gender. These data were collected on the Thursday, Friday, and Saturday of five consecutive weeks in July and August of 2016, counting passengers on one route each week. Conducting the tally from Thursday through Saturday allowed for observation of ridership for a typical weekday, a transition day (Friday), and a weekend day.

An advantage of on-board surveys is that respondents do not need to remember a past trip; they are asked about their travel while they are traveling. A limitation, however, is that this approach cannot provide information about commuter van use by a representative sample of neighborhood residents. As with all surveys, nonresponse bias—where those who choose to answer the survey are systematically different from those who choose to decline the survey—was a concern. Because this survey was administered in person, the researcher directly observed differences between respondents and non-respondents. Specifically, younger commuter van riders were more likely to respond to the survey than older riders.

The survey included questions about MTA transit use, commuter van-related travel choices, the current commuter van trip, and basic demographics. Data collected in the field were cleaned, coded, and analyzed using Microsoft Excel and Stata 14 software. Goldwyn [6] also used on-board surveys in his study of the Brooklyn commuter vans—the only existing study of New York's commuter vans that is comparable to ours. Many of our survey questions overlap with those asked in Goldwyn's 2013 survey of Brooklyn commuter van riders [6].

Table 1 provides summary statistics for the Eastern Queens survey as well as a direct comparison between the two van rider survey datasets. This comparison is important because it highlights the parallels between the van riders' characteristics in Brooklyn and Eastern Queens, which may speak to the success of commuter vans in these neighborhoods. Data from the American Community Survey (2016) show that approximately 55% of the population in the study area were female, 50% were immigrants who commuted to work, 43% commute to work via transit, 60% of the households reported incomes over $50,000, and 81% of households owned at least one car. A comparison with these census data indicates that van riders reflect the study area population in their immigrant status, but are notably poorer than the general population, less likely to own a car, and more likely to ride transit. Although these differences between van riders and the study area population likely reflect reality, they may be exaggerated by the fact that our survey respondents were biased toward younger commuter van riders. To the extent that younger adults have relatively lower incomes, are less likely

to own a car, and are more likely to ride transit than older adults, our survey results may also be biased in this way.

**Table 1.** A Comparison of Surveys.

| Percent of Commuter Van Riders Who: | 2016 Eastern Queens Commuter Van Survey (*N* = 376) | 2013 Brooklyn Commuter Van Survey [6] (*N* = 198) |
|---|---|---|
| Are Female | 59% | 63% |
| Are Immigrants | 50% | 56% |
| Own a Car | 26% | 30% |
| Report Household Incomes Above $50,000/Wage Above $17 * | **21%** | **43%** |
| Regularly Use MTA Transit | **74%** | **84%** |
| Have a MetroCard | **63%** | **76%** |
| Have an Unlimited MetroCard | 33% | 31% |
| Regularly Ride Vans at Least Once a Week | 90% | n/a |
| Ride Vans Because They: | | |
|     Are Fast | **62%** | **80%** |
|     Are Convenient | **39%** | **50%** |
|     Are Cheap | 52% | 45% |
|     Have Short Wait Times | 39% | 39% |
| Will Transfer to MTA Transit on This Trip | 53% | n/a |
| Are Going Home on This Trip | 18% | n/a |
| Are Going to Work on This Trip | 30% | n/a |
| Are Going to School on This Trip | 23% | n/a |
| Are Going Shopping on This Trip | 15% | n/a |

Note: Boldface font and shading are used to indicate statistical significance of differences in proportions between the two surveys. * The Eastern Queens survey reports household incomes, while the Brooklyn survey reports individual hourly wages. To convert these to (roughly) comparable metrics, we calculated the percentage of survey respondents reporting incomes/wages above the approximate median household income and hourly wage in New York City at the time of each survey. In both surveys, some respondents refused to provide income information, so the number of valid responses to this question was 324 for the Eastern Queens survey and 146 for the Brooklyn survey.

Hypothesis tests indicate a statistically significant difference between the two van rider surveys for the percentages shown in boldface and orange shading. There are many similarities between the van riding populations in these two neighborhoods. The gender balance among riders, the fraction of riders who are immigrants, and the percent who own cars are not statistically different between these two samples. The fraction of riders who have MetroCards at all—the MTA official fare payment card—and use MTA transit was significantly lower in Eastern Queens. This is not surprising because Eastern Queens is located on the edge of New York City, whereas the Brooklyn van routes are more centrally located. There were also significant differences in the fraction of van riders who responded that van speed and convenience were key reasons why they chose the vans over MTA transit. This may reflect real differences in relative speeds between vans and MTA buses in these different neighborhoods.

## 5. Results

*5.1. How Many People Ride Commuter Vans in Eastern Queens?*

Table 2 presents ridership tally results on five commuter van routes in Eastern Queens. The totals represent the daily ridership of one commuter van excluding the researcher; the percentages of males and females are provided as well. The capacity of the commuter vans on four of the routes was 13 seated passengers, and that of the Far Rockaway route was between 20 and 25 seated passengers.

**Table 2.** Commuter van male and female ridership tally results on individual routes.

| Van Routes | | Thursday | Friday | Saturday | Average |
|---|---|---|---|---|---|
| Rosedale | Male | 31% | 41% | 45% | 39% |
| | Female | 69% | 59% | 55% | 61% |
| | Total | 163 | 178 | 144 | 162 |
| Far Rockaway | Male | 46% | 41% | 42% | 43% |
| | Female | 54% | 59% | 58% | 57% |
| | Total | 190 | 247 | 212 | 216 |
| Linden | Male | 48% | 34% | 37% | 40% |
| | Female | 52% | 66% | 63% | 60% |
| | Total | 232 | 104 | 161 | 166 |
| Green Acres Mall | Male | 36% | 44% | 37% | 39% |
| | Female | 64% | 56% | 63% | 61% |
| | Total | 139 | 124 | 149 | 137 |
| Cambria Heights | Male | 44% | 44% | 39% | 42% |
| | Female | 56% | 56% | 61% | 58% |
| | Total | 116 | 142 | 126 | 128 |

Average ridership across all commuter van routes in the study area was 161 people per day per van in summer 2016. Interestingly, daily ridership was not appreciably different between weekdays and weekend days. All commuter vans had higher female than male ridership—consistent with our survey results—and this pattern was observed throughout the day. Transit safety has been shown to be a defining factor for women's transport mode choice [26]; high female ridership could be evidence that commuter vans are perceived as a safe mode of transit in Eastern Queens. In fact, survey respondents in this study selected comfort and safety as some of the positive traits of the Eastern Queens commuter vans.

We used these tally numbers to estimate the total daily commuter van ridership in Eastern Queens in the summer by simply multiplying by the number of vans on the streets. According to the President of the New York Commuter Van Association, there were approximately 270 registered commuter vans and 70 unregistered vans in the study area [25]. The total daily summer commuter van ridership in the study area was therefore approximately 55,000. This tally may actually undercount annual van ridership. During fieldwork, we heard from drivers that ridership is seasonal, and that van ridership is highest in winter. This is both because people prefer not to stand in the cold waiting for the bus and because the vans can maneuver better than buses in snowy conditions.

Putting our estimate of 55,000 van riders per day in context, MTA transit's reported bus ridership for 2016 was approximately 53,000 riders each weekday on the five parallel bus routes [27,28]. The commuter vans, then, are serving about half of the transit trips along these routes.

*5.2. What Are the Socioeconomic Characteristics of Commuter Van Riders?*

The Eastern Queens van rider survey collected four demographic characteristics of respondents: household income, gender, immigrant status, and age category. Table 1 reports summary statistics on the first three of these variables, illustrating that van riders are relatively low-income, include more women than men, and are about half immigrants.

Table 3 provides a look at how these demographic characteristics are associated with four key transportation choices that were also reported by survey respondents: whether a car was available for the trip they were making on the commuter van at the time of the survey, possession of an unlimited MetroCard, frequency of commuter van use, and whether their trip at the time of the survey would have been free on MTA transit. The last of these occurs when van riders either possess an MTA unlimited-ride MetroCard, or they are planning to transfer to an MTA bus or subway (MTA transfers are free within a 2-hour time window). Chi-squared independence tests were performed to

identify the statistical significance of the differences between each demographic category and transport choice, and Table 3 provides these test statistics for each pair of variables. Statistically significant differences are indicated with asterisks, and those that meet standard statistical significance thresholds (*p*-value < 0.05) are highlighted in bold font and orange table shading.

**Table 3.** Associations between demographics and respondent transport choices.

| | | Car Available for This Trip | | Unlimited MetroCard | | Use Van More Than 1x/Week | | Paying $2 Van Premium on This Trip | |
|---|---|---|---|---|---|---|---|---|---|
| | | Pct. | $\chi^2$, N | Pct. | $\chi^2$, N | Pct. | $\chi^2$, N | Pct. | $\chi^2$, N |
| Annual Income | Under $50k | **15%** | **21.96***** | 33% | 0.02 | 86% | 0.90 | 64% | 0.10 |
| | $50k or Greater | **42%** | **N=313** | 34% | N=321 | 81% | N=324 | 66% | N=324 |
| Gender | Male | **30%** | **7.93***** | **39%** | **4.10**** | 82% | 0.81 | 66% | 0.52 |
| | Female | **17%** | **N=330** | **29%** | **N=338** | 86% | N=341 | 62% | N=341 |
| Age | 18 – 30 | **14%** | **13.10***** **N=325** | 35% | 5.12* N=333 | 87% | 3.18 N=336 | 60% | 2.12 N=336 |
| | 31 – 45 | **26%** | | 37% | | 84% | | 69% | |
| | Over 45 | **37%** | | 21% | | 77% | | 63% | |
| Immigrant | Yes | 25% | 1.70 | 31% | 0.61 | 85% | 0.56 | 65% | 0.21 |
| | No | 19% | N=317 | 36% | N=325 | 82% | N=328 | 63% | N=328 |

Note: Asterisks, boldface font, and shading used to indicate statistical significance of $\chi^2$ tests. *** Indicates *p*-value < 0.01, ** Indicates *p*-value between 0.01 and 0.05, * Indicates *p*-value between 0.05 and 0.10. Darker shading and boldface font indicate standard statistical significance with *p*-values < 0.05. Light shading indicates marginal statistical significance with *p*-values between 0.05 and 0.10.

The main finding from this analysis is that among van riders, these demographic characteristics are not significantly associated with frequency of van use or paying a premium to ride the vans. Household income, gender, and age are, however, associated with car access; higher-income, older, and male van riders were more likely to have car access. Gender was also associated with possession of an unlimited MetroCard, with men more likely to have the MTA passes than women.

*5.3. Why Do Riders Choose the Commuter Vans over Public Buses?*

Existing research on jitney-style transport services provides a variety of reasons why people might choose to ride commuter vans. Cervero and Golub [29] describe these services as highly flexible. King and Goldwyn [1] and Goldwyn [6] attribute the success of the New York dollar vans to time savings, and low cost is often cited as an important characteristic of jitney-style transportation services [29,30].

In Eastern Queens, the van riding experience differs from the bus riding experience. Commuter vans are much smaller than MTA buses. In part because it is difficult to stand in a van, van riders are almost always comfortably seated, whereas "standing room only" is common on peak-hour buses. Buses pick up passengers only at designated stops, while vans pick up riders anywhere along their routes, as long as there is an available seat. Buses do not deviate from their assigned routes, while vans will go off-route to provide more direct service to individual passengers. In the course of conducting fieldwork for this study, we observed several instances when the van drivers went off the regular route to drop off kids at a daycare or drop off older passengers and those with shopping bags or heavier loads at their doorsteps. Sometimes the drivers charged an extra dollar or two for the service, but most commonly this was done simply as a favor to regular van customers. Once, a passenger requested to be taken to the Long Island Railroad station several miles from the last van stop and the driver charged $5 for the ride—less than a taxi or ride hailing service would have cost.

The Eastern Queens van rider survey asked the question, "Why do you take dollar vans?", with the opportunity for respondents to choose multiple answers. Table 1 reports that 62% of Eastern Queens survey respondents cited speed of travel as one reason that they chose the commuter vans over the MTA bus, and 39% also reported that short wait times were an attractive feature of the vans.

To investigate the extent to which the vans are actually faster than the city buses in Eastern Queens, we compared commuter van and bus trip times for a trip from one end of each commuter van route to the other. Note that this comparison provides the maximum possible time savings provided by the vans because some riders do not ride the entire route. Van riders, however, often do ride the full length of the route—especially at peak hours.

During the tally exercise, the time spent on every trip was recorded. Separately, time spent on MTA bus trips with the same origin and destination was recorded, although the routes followed were not strictly identical. Metropolitan Transportation Authority bus ride times were recorded only once through a round trip. Table 4 reports the average van ride duration during the tally, the observed MTA bus trip duration, and the official MTA scheduled bus trip duration. Commuter van trips on all routes in the study area are faster than trips on MTA buses along similar routes, and for some routes the difference is large. When travelers choose to use a commuter van, then, they are buying time.

**Table 4.** Travel time comparison between commuter vans and city buses.

| City Buses | Commuter Van Average Time (Minutes) | Observed MTA Bus Trip Time (Minutes) | Official MTA Scheduled Trip Time (Minutes) | Commuter Vans on Parallel Routes |
|---|---|---|---|---|
| Q5 | 29 | 49 | 42 | Green Acres Mall |
| Q83 | 22 | 27 | 31 | Cambria Heights |
| Q4 | 23 | 31 | 32 | Linden |
| Q111 | 27 | 40 | 43 | Rosedale |
| Q114, Q113 | 42 | 55 | 68 | Far Rockaway |

As indicated by the survey results, commuter van riders value time savings in both reduced ride times and reduced wait times. We did not record the difference in wait times to provide a quantitative comparison, but researcher observations revealed that wait times for vans in Eastern Queens are usually shorter than wait times for the MTA buses. The reasons for the shorter van wait times are structural. During peak times, demand for both buses and vans is high. The bus and van routes do not begin at the same location, however, and often the buses are full even before they reach the van route terminus. This means that peak van wait times at their terminus are clearly shorter than bus wait times; the vans begin empty there, but passengers may have to wait for multiple MTA buses to pass before one with available space arrives. At off-peak times, the bus headways are longer but the van frequency remains high throughout the day. In addition, the fact that commuter vans are not bound by strict schedules can often reduce their wait times.

Low cost was reported to be an attractive commuter van trait for 52% of van rider survey respondents in Eastern Queens. For cost-sensitive residents who travel locally, the cost advantage of commuter vans ($2 versus $2.75 for MTA buses in the summer of 2016) is indeed significant. As noted earlier, however, many van riders actually pay a $2 *premium* to ride the vans because their rides on MTA buses are effectively free—they have already paid for them.

We find that *at least 63%* of the Eastern Queens survey respondents are in this category, paying a $2 premium for their van rides. We cannot report the exact percentage due to an unfortunate oversight in survey design. The Eastern Queens commuter van rider survey asked whether respondents were going to transfer *to* MTA bus or subway transit from their commuter van ride, but did not ask whether respondents had transferred *from* an MTA bus or subway to the commuter van. Sixty-three percent of our respondents either had an unlimited-ride MetroCard or were planning to transfer to MTA transit to complete their trips. We expect that additional respondents had transferred from MTA transit to the commuter vans, but we do not know how many. For this reason, we report that at least 63% of the survey respondents were paying a premium for their van rides.

This finding is remarkable, especially when considering the low-income profile of most van riders. Table 3 indicates that a binary measure of income (over or under $50,000 annually) is not related to whether a respondent paid a premium for his or her van ride. A more detailed look at the data reveal

that at least 64% of riders in the lowest income category (less than $20,000 annual income) paid a van premium, and at least 71% of riders in the second-lowest income category ($20,000–$30,000 annual income) paid a van premium. We explored the choice to pay a van premium with multivariate analysis, and found that paying a premium to ride the vans was not significantly explained by any of the rider characteristics that we collected in our survey. This result suggests that regardless of income and other demographic characteristics, van riders value the service quality of the commuter vans over the MTA buses enough to pay an extra $2 to ride them.

We looked at the difference in the reporting of cost as a reason for riding the vans between those survey respondents who would and would not save money on their current van trip (see Table 5). We did find a difference, but it was surprisingly small and not statistically significant. Fifty-seven percent of van riders who would save money on their trip by using the vans reported the low cost of the vans as a reason they chose the van over the MTA bus, but this figure for riders who were actually paying a premium for their van ride remained high at 49%. Perhaps these riders considered $2 to be inexpensive for the premium in service that they receive when riding in the vans compared with the city bus; they arrive faster and are perhaps more likely to get a seat. It may be, however, that these riders were simply thinking that the base fare for the vans is lower than that of MTA transit—even though for their particular ride at the time of the survey, MTA transit would have been effectively free.

**Table 5.** Associations between respondent reasons for riding vans and transport choices.

|  | Fast | | Convenient | | Cheap | | Short Wait | |
|---|---|---|---|---|---|---|---|---|
|  | Pct. | $\chi^2$, N | Pct. | $\chi^2$, N | Pct. | $\chi^2$, N | Pct. | $\chi^2$, N |
| Car Available | 61% | 0.04 | 44% | 0.80 | **42%** | **3.98**\*\* | 38% | 0.08 |
| Car Not Available | 62% | N=358 | 39% | N=358 | **54%** | **N=358** | 40% | N=358 |
| Unlimited MetroCard | **70%** | **4.30**\*\* | 43% | 1.42 | 46% | 3.13\* | 42% | 0.44 |
| No Unlimited MetroCard | **58%** | **N=368** | 37% | N=368 | 56% | N=368 | 38% | N=368 |
| Use Van 1x/Week or Less | **49%** | **4.91**\*\* | 41% | 0.05 | 42% | 2.62\* | 36% | 0.36 |
| Use Van More Than 1x/Week | **64%** | **N=371** | 39% | N=371 | 54% | N=371 | 40% | N=371 |
| Paying $2 Premium | 64% | 1.17 | 41% | 1.21 | 49% | 1.89 | 43% | 3.64\* |
| Saving Money | 58% | N=371 | 36% | N=371 | 57% | N=371 | 33% | N=371 |

Note: Asterisks, boldface font, and shading used to indicate statistical significance of $\chi^2$ tests. \*\*\* Indicates *p*-value < 0.01, \*\* Indicates *p*-value between 0.01 and 0.05, \* Indicates *p*-value between 0.05 and 0.10. Darker shading and boldface font indicate standard statistical significance with *p*-values < 0.05. Light shading indicates marginal statistical significance with I-values between 0.05 and 0.10.

Tables 5 and 6 illustrate additional associations between transport choices of van riders and their reasons for riding the vans, and between the different transport choices made by van riders. Those who have unlimited MetroCards overwhelmingly list speed as a key reason that they use the commuter vans, and are less likely than those without unlimited MetroCards to specify the vans' low cost as a motivating factor. This makes sense, as these riders are among those paying a $2 premium to ride the vans. Those who use the vans more than once a week are more likely to list both speed and low cost as reasons that they use the vans, compared with those who use them once per week or less. Higher frequency van riders were also less likely to own a car or have an unlimited-ride MetroCard.

**Table 6.** Associations between respondent transport choices.

| | Car Available for This Trip | | Unlimited MetroCard | | Use Van More Than 1x/Week | |
|---|---|---|---|---|---|---|
| | Pct. | $\chi^2$, N | Pct. | $\chi^2$, N | Pct. | $\chi^2$, N |
| Unlimited MetroCard | 19% | 1.27 | | | | |
| No Unlimited MetroCard | 24% | N=359 | | | | |
| Use Van 1x/Week or Less | **37%** | **9.02*** | **49%** | **7.84*** | | |
| Use Van More Than 1x/Week | **19%** | **N=362** | **30%** | **N=372** | | |
| Paying $2 Premium | 21% | 0.72 | n/a | n/a | 84% | 0.07 |
| Saving Money | 25% | N=362 | n/a | n/a | 85% | N=376 |

Note: Asterisks, boldface font, and shading used to indicate statistical significance of $\chi^2$ tests. *** Indicates *p*-value < 0.01, ** Indicates *p*-value between 0.01 and 0.05, * Indicates *p*-value between 0.05 and 0.10. Darker shading and boldface font indicate standard statistical significance with *p*-values < 0.05. Light shading indicates marginal statistical significance with *p*-values between 0.05 and 0.10.

### 5.4. Do Vans Complement or Compete against MTA Transit Service?

Literature has shown that private jitney-style services can compete against formal transit, complement formal transit, or, in cases where formal transit is absent, substitute for formal transit [4,6,31]. Both King [4] and Goldwyn [6] argue strongly that Brooklyn's commuter vans serve as important complements to, occasional substitutes for, and minor competitors to formal transit. Our research provides evidence that this is also true in Eastern Queens.

An important feature of the commuter van rider population is regular use of multiple transport modes. The van rider survey asked riders to identify the transport modes they regularly use, and more than three-quarters of the respondents checked the box next to two or more modes. The survey also included a question about how respondents would make their current trip if the vans were not available. Of the 368 survey respondents who answered this question, only seven of them (under 2%) responded that they would not make the trip. The other 98% identified at least one alternative mode that they could use. In particular, 78% of commuter van riders identified the MTA transit bus as an alternative mode for the trip they were taking. This provides some evidence that the commuter vans compete directly with neighborhood MTA buses for riders.

Nonetheless, the net financial impact of the commuter vans on the MTA is almost certainly positive. As mentioned earlier, we estimate that commuter vans in Eastern Queens serve approximately the same number of trips as the buses along parallel routes. If all of the van riders moved over to the buses, then, the MTA would need to accommodate a doubling of ridership on routes whose buses are currently full (i.e., standing room only, and sometimes even standing room is not available) at peak hours. This would be expensive. Importantly, in doing so, the MTA would receive less than a 40% increase in fare revenue. This is because more than 60% of commuter van riders would not have paid extra to ride the MTA buses; they either held an unlimited-ride MetroCard or they were transferring to the MTA for the next leg of their trip.

We conclude that the Eastern Queens commuter vans complement MTA transit service in two important ways. First, the vans serve as a high-quality feeder route system for the MTA's subway, commuter rail, and express buses that connect Jamaica Center to other parts of the city and region. Second, the vans relieve crowding on Eastern Queens MTA buses during peak hours, when buses are often full. From the MTA's point of view, this should be a win–win–win situation. The MTA gets most of the fare revenue while providing only half of the feeder bus service along these relatively low-density routes, the passengers on the vans enjoy (and are willing to pay for) the relatively fast van service, and the commuter van drivers and owners are able to make a living.

In addition, the role of commuter vans as occasional substitutes to MTA transit during harsh weather is apparent. For instance, even with New York City's transportation system largely shut down after Hurricane Sandy, privately operated "dollar vans" stayed in operation [9]. An MTA bus driver quoted by Olean [32] said "[The vans] help us out in a lot of ways, when the weather gets bad or when we're overwhelmed during rush hour, they take [away] some of the burden."

## 6. Discussion

Commuter vans are a rare example in the US context of a successful and sizable private collective transport service. The industry has been operating for over 30 years, and along the Eastern Queens van routes, our estimates suggest that the vans carry as many passengers as the MTA buses. This research provides an initial picture of the Eastern Queens commuter vans and their riders, based on field research conducted during the summer of 2016. We aimed to estimate van ridership, as well as identify who rides the vans and why they do so despite the existing parallel MTA bus service. The answers to these questions provide insight into the extent to which the commuter vans complement or compete with MTA transit in this neighborhood—an important policy question, especially in the current era of rapidly proliferating mobility services.

Commonly-held views—especially by those who do not use the commuter vans—are that the vans are a cheaper and therefore inferior transport service compared to the MTA buses. It is true that the base fare is lower on the vans—$2.00 versus $2.75 for a single ride in the summer of 2016. The fare story is complicated by the fact that MTA offers free transfers between their buses and subways, as well as unlimited ride passes and further discounts for children and seniors. Van fares do not vary, van passes do not exist, and there are no transfers included in the van base fare. One of the most remarkable findings from this study was that at least 63% of the van riders we surveyed were paying a $2 premium to use the vans; their ride for the trip they were taking at the time of the survey would have been free on an MTA bus. Most commuter van riders are relatively low-income, and the fact that many of them are paying extra to ride the vans shows that even poor riders are willing and able to pay for higher transport service quality.

Indeed, our data indicate that vans actually do provide a premium service. Most importantly, they are substantially faster than MTA buses because they do not adhere to a schedule or stop at pre-designated locations. Our data show that Eastern Queens vans are 28–38% faster than the MTA buses for a trip that traverses the entire van route, corresponding to a potential time savings of between 9 and 26 minutes. Earlier work by Goldwyn [6] estimated that vans along Flatbush Avenue in Brooklyn were 50% faster than parallel MTA buses. Vans also tend to have shorter wait times than the buses, mainly because there are more of them on the streets, so their frequency is higher.

Although the vans clearly siphon riders away from MTA buses, our work suggests that the net financial impact on the MTA is likely positive. The conclusion that commuter van-type services complement formal public transit is not unique to New York. Cervero [30] pointed out the complementary role played by jitney-style transport services in many cities around the world, and the private commuter shuttles of San Francisco provide an example in yet another context. The executive director of the San Francisco County Transportation Authority cited the importance of these shuttles, admitting that "at the end of the day, [our formal transit] can't meet the demand" [19].

*Parallels between Commuter Vans and Other Private Mobility Services*

The New York commuter van industry has shown resilience and adaptability throughout multiple phases of regulation over decades. In their first decade of operation, the government took a laissez faire approach, enabling the development of a market-driven business model to provide flexible, adaptable, and affordable transport service. When TLC took over commuter van regulation in 1993, they implemented regulations that limit commuter vans' operations flexibility and adaptability—fundamental components of their business model. This mismatch is attributed by Goldwyn [6] (p. 31) to "a profound lack of communication and coordination between dollar van interests and local officials", stemming from TLC naivety about the vans they regulate. When the New York City Council discussed the commuter vans as recently as 2017, one councilman argued for increased regulation, while another argued for regulatory reforms that would better support the commuter van industry [33].

This history of commuter van interaction with regulators bears a striking resemblance to the current interactions playing out between cities and private ride hailing services like Uber and Lyft;

cities are now working toward regulating these services after a few years of largely unconstrained operations. There is a recognition that these services fill a gap while at the same time a recognition that they may have negative impacts on public transit, congestion, and safety. In a city like New York, then, should the government seek to improve their transit system to fill the gaps that allow commuter vans, Uber, and Lyft to thrive, or should it diversify transit to explicitly partner with private services? While we do not answer this question, our evidence can help inform the ongoing debate on whether the "positive" roles outweigh the negative ones for commuter vans.

It is important to note that New York is unique in the US in terms of its combination of a well-developed public transit system, high population density, high car parking costs in commercial districts, and high levels of traffic congestion. Together, these characteristics effectively encourage New Yorkers to use transit—even if they have to transfer to a train from a feeder bus, van, or even ride hail or taxi. Some lessons from this research, therefore, may not apply to cities that do not have these characteristics.

## 7. Conclusions

The commuter vans of New York and similar services elsewhere serve a purpose that is important for transportation planners to understand. Privately-owned shared modes are growing, and public entities should identify opportunities to engage with them to ensure that benefits are widely and equitably shared [13]. In the current regulatory environment in New York, the characteristics that make commuter vans attractive are out of reach for MTA transit—they provide a premium service at an affordable price without government subsidies. Although our data illustrate that commuter vans in Eastern Queens likely have a positive net financial impact on the MTA, this conclusion is highly context-dependent. As privately-owned shared transport modes proliferate across the globe, continued research is needed to develop context-specific strategies to integrate them with formal transit for the mutual benefit of cities, entrepreneurs, and most importantly, commuters.

**Author Contributions:** Individual contributions to this article are as follows: conceptualization, C.M. and D.S.; methodology, C.M. and D.S.; fieldwork, C.M.; data curation, C.M.; Writing—Original Draft preparation, review, and editing, C.M. and D.S.; supervision, D.S.; funding acquisition, C.M.

**Funding:** This research was funded by the Fulbright Scholar program. Additional travel and research expenses were funded through a small grant from Arizona State University.

**Conflicts of Interest:** The authors declare no conflict of interest.

## Appendix A  Commuter Van Rider Survey

1.　What is your regular mode of transportation? Check all that apply. (N=375)

☐　　Personal Vehicle (71)
☐　　Bus (214)
☐　　Subway (202)
☐　　Dollar van (316)
☐　　Bicycle (36)
☐　　Walking (38)
☐　　Other. Please specify (1) _________________________________

2.　Do you have an MTA MetroCard? (N=373)

☐　　Yes, I have the 30-day unlimited ride MetroCard (71)
☐　　Yes, I have the 7-day unlimited ride MetroCard (46)
☐　　Yes, I have the pay-per-ride MetroCard (111)
☐　　I have the 7-day express bus pass (7)
☐　　No, I pay in cash (97)

☐      No, I do not have a MetroCard (29)

☐      No, I do not take the subway or city bus (13)

3.   How often do you take dollar vans? (N=364)

☐ Every day (51)                   ☐ Once a week (24)
☐ 6 days a week (45)            ☐ Twice a month (14)
☐ 5 days a week (137)          ☐ Once a month (7)
☐ 4 days a week (32)            ☐ Less than once a month (13)
☐ 3 days a week (25)            ☐ This is my first trip on a dollar van (1)
☐ 2 days a week (16)

4.   Where are you going now? (N=362)

☐      Home (66)

☐      School (including College/University) (84)

☐      Work (107)

☐      Shopping (53)

☐      Social/Church/Personal (48)

☐      Other (please specify) (4) _________________________________

5.   Do you have a car that you could have used to make this trip? (N=362)

☐      Yes (81)

☐      No (281)

6.   Will you transfer to another bus or train on this trip to where you are going now? (N=360)

☐      Yes, I will transfer to the city bus (60)

☐      Yes, I will transfer to the subway (106)

☐      Yes, I will transfer to another (Please specify) (25)

☐      No (171)

7.   If dollar vans were not available, how would you make this trip? (N=368)

☐      Personal vehicle (49)

☐      City bus (286)

☐      Subway (73)

☐      Bicycle (21)

☐      Walk (15)

☐      Not take the trip (9)

8.   Why do you take dollar vans (Check all that apply)? (N= 371)

☐ Flexible van route (72)         ☐ Short wait time (145)
☐ Fast (230)                     ☐ I know the driver (20)
☐ Comfortable (59)             ☐ Everyone rides dollar vans (2)
☐ Cheap (193)                  ☐ My only option (0)
☐ Safe (34)                      ☐ Other (0)
☐ Convenient (102)            Please specify_________________

9.   Do you think dollar vans are important in your neighborhood? (N=365)

☐      Yes (338)

☐      No (5)

☐      I don't know (22)

10.   In your opinion, what makes informal transportation thrive in your neighborhood?

_____________________________________________________________________________________

11. What is your gender? (N=341)

    ☐ Male (139)
    ☐ Female (202)

12. What is your age? (N = 336)

    ☐ 18-30 (150)
    ☐ 31-45 (126)
    ☐ 46-60 (52)
    ☐ 61 or more (8)

13. Which option below best describes your total annual household income? (325)

    ☐ Under $20,000 (83)
    ☐ $20,000 – $29,999 (69)
    ☐ $30,000 – $39,999 (30)
    ☐ $40,000 – $49,999 (75)
    ☐ $50,000 – $59,999 (23)
    ☐ $60,000 – $69,999 (24)
    ☐ $70,000 – $79,999 (18)
    ☐ $80,000 or greater (4)

14. Informal public transportation has been shown in previous studies to be prevalent in immigrant communities in the US By answering this question you help test this conclusion. Are you an immigrant? (330)

    ☐ Yes (166)
    ☐ No (166)

15. If so, which country did you come from? ___________________________________

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
