# Peer review of "Do Private Transport Services Complement or Compete against Public Transit? Evidence from the Commuter Vans in Eastern Queens, New York"

_urbansci, doi:10.3390/urbansci3010024_

Reviewer 1 Report

This is a very well-written and researched paper. The objectives, methods, and results are clear. The only improvement that I see needed is adding something about generalizability to other regions or systems. This seems to be a very specific case study, so some comment about the usefulness and applicability of the results would be helpful.

Author Response

Please find attached a Word document that provides an overall summary of our revisions followed by point-by-point responses to all reviewer comments (including yours).

Reviewer 2 Report

This paper provides an initial picture of the ridership and riders of commute vans based on field research in Eastern Queens, New York. The paper is clearly structured and written. It is also seen the effort that the researcher made to collect the data. However, because of several deficiencies and open questions, this objective has not been fully achieved. It is therefore recommended that the paper be accepted subject to a major revision.

General comments:

·         The main aim of the article is trying to answer the question on “do commuter vans complement or compete against MTA transit service”. But the results and the analysis of the paper seem not answer it explicitly.  

The authors have stated the two modes (i.e., commuter van and transit bus) are both complement and compete. It is suggested that the authors should explicate the certain

circumstances which the two modes are complementary or competitive. For example, the authors should present a comparison of the two modes with more features, like working schedule, headway (peak hour and non-peak hour), tariff, vehicle capacity, average travel time (peak/non-peak hour), number of stops, etc.? Also qualitative comparison of both advantages and shortages of the two modes, like safety, comfortability (all have seats, occupied space), flexibility, etc.?   

·         One of the key findings of the paper is that there is almost half of van riders like to pay extra fee to use the vans beside the bus ticket they have already paid. It would be better to make an analysis on travel affordability regarding the household income. It may be interesting to know to which extend passengers willing to pay to use commuter van or other ridehailing services.

·         More policy recommendation should be discussed for PT operators and other ridehailing service providers.

Specific comments:

Line 80, six or five main routes. The following contents only mentioned five routes.

Line 94, add “annual” before income.

Figure 1, add the MTA transit lines into the map to comparing with the commuter van lines.

Line 108, it mentioned “56% of commuters in the study area used public transportation, 23% carpooled, and 31% drove to work in 2015”, which is not consistent with the estimation the authors made for commuter vans.  In line 289-290, “the commuter vans, then, are serving about half of the transit trips along these routes”. The estimation is correct?

Section 3, the paragraph from line 149 to 155 should be moved to line 118 in order to present the whole picture of the survey conduction.

Table 1, add average journal time, average trip distance if they are available.

Line 203, the authors mentioned the MTA bus ride time were recorded once through a round trip. Why do not collect them through Google or other similar APPs? It has very detailed traffic data by hour, isn´t?

Table 3, the title should be “travel time”. And it is suggested that the authors should place the unit on the top along with the variables.

Line 309, “smaller” is not the appropriate word, “less capacity” maybe?

Author Response

(The authors gave the same response as above.)

Reviewer 3 Report

This is very interesting research with potential. However, the authors are unsure of their perspective – a major problem I attribute to the lack of context. The paper would need to be completely reworked before publication. The authors need to spend time explaining why the research is important and why the particular research questions chosen are important. The paper needs theoretical context to be understood.

None of the research questions are well answered by the authors. In my opinion, this is because the authors do not appear to understand why this research may or may not be important. The authors state that it is because Uber and Lyft are coming with similar services. But this is a regulated commercial service already. The addition of a competitor to the market is not theoretically or practically interesting.

Why did they choose to answer these four research questions? (And question two is never answer or discussed.) How do the questions relate to one another or to a theory or policy? This paper lacks the context that would enable readers and authors to understand the results. The lack of context leads the authors to draw conclusions that are not supported by the data. There are important transportation economic and transportation equity policy questions that could be discussed. The research has potential, but the authors do not tell me why. 

More detailed notes:

Abstract

The answers at the end of the abstract do not answer the question put out at the beginning. Are we wondering if vans steal passengers or are we wondering if they are better for the riders who use them? 

Context

There is no context. I have lots of questions, especially since there are policy implications touted. Context would help me and the authors understand the research and its implications. There is no way to put these numbers in perspective now.

How are vans regulated? By whom? How strictly? 

What kinds of rules seem to indicate the different attitudes between NJ and NY perspectives? Or can we tell by public statements? 

Who owns the vans? Are they big companies or individual entrepreneurs?

Do these occur in other markets in the US? They do not have to for the research to be interesting, but I am wondering. (I see later that they are in San Francisco. Is there research there that is interesting?)

Vans are common in developing countries. And may exist in other developed countries as well. What does research there tell us about interpreting these results? Does research find vans steal customers from transit? Are the important complements? Are they faster? More reliable?

You do not have to really answer all of these questions. But you can see that I am thrashing around to put the results in some kind of context. Why is this paper theoretically or practically important?

 Methodology

Fine survey design overall. 

Perhaps too many details in this section. I would like a shorter methodology section so that you can have some context. 

Can you do cross tabs and tell me more about what is going on in the data?

Are there any analyses that you can do beyond descriptive that might set up some causation hypotheses for future research? You can’t demonstrate causation, but you can start to point in some directions. This would make the discussion and conclusion more interesting. 

Results

Why are the raw results for each question not presented? In the appendix is okay. Or just describe the questions you use in the particular paper. 

4.1. An understanding about the importance of gender is missing from the background and so I do not know why we chose to discuss this at length. The only thing I get is that these vans are perceived to be safe. But was that ever an issue if they are overseen by the city? It would be more interesting if it related to the kinds of jobs they had or where they lived or something. Tell me why the gender breakdown is important. 

At the end, you bury two reasons why the vans might be used, but do not discuss them in the appropriate section (4.2). 

4.2. Time savings section is good, although the section about the time spent is confusing. Given the language, I was expecting to see time spent per rider, but it is not presented. The point that is made is fine, just make it a bit easier to understand. 

 4.2 The authors seem mystified by the fact that riders would pay a $2 premium. If it is faster and more reliable in bad weather, then I do not see that this finding is remarkable. This convenience premium is what Uber and Lyft are betting on. It would be a much more interesting finding if those paying the premium were richer. Does this help poor people? Or is it just a pre-Uber skimming of wealthier customers?

 4.3 There is no real answer presented to this question. The background data about NYC official perspective should be presented at the front. (And it is confusing because you state that these have to be pre-arranged here, but from the earlier description, I thought they could be just met at certain stops or hailed like taxis.) A policy position, which may not be related to the question, is not the answer as to whether they compete. The anti-van rules could be in place because of fears of congestion or how they look or because they are not the mayor’s buddies or other issues not relevant to the paper. The rules might be meant to reduce potential for competition, but you have to demonstrate that.

4.3 More importantly, I’m not sure there is an operational distinction between complete and complement. Compete implies that they are taking riders from the buses, but you report that the buses are often full. So they would complement a stressed service. It is why people pay a premium. It would be interesting if the bus agency has data of who rides this route. It would be very interesting if the vans took a richer or poorer or particular kind of rider away. That would be a more interesting finding. 

4.3 In sum, I think the question is important, but the authors need to rethink whether and how they can answer it. 

** Research question two is completely missing from the results. The answer to the socioeconomic question – and crosstabs using it – would make the paper much more interesting. 

Discussion

Paragraph starting at line 286. This paragraph is a rehash of the results and then a drawing of the conclusion that they are competing with MTA buses. Where does that come from? Although I do not agree with the difference between competing and complemenary, the authors end section 4.3 with an example that they try to indicate it is complementary. So the statement contradicts their own findings section.  

Same paragraph. Why do the authors think this is surprising or alarming? What context has been provided that anyone cares? This is exactly why the presentation of literature and theory is needed. 

Page 10, line 293. Whose commonly held views? If there is evidence of this, why is it important? In this same paragraph, the authors veer into complicating the fare story, but with no indication of why it matters. Why do the variety of passes complicate the story? What are you trying to tell me? Then the paragraph concludes with a third new idea, the stunning fact that people pay a premium when they have not built a case that this is all that compelling a finding. 

Line 303. The authors explain the stunning fact from the previous paragraph in this paragraph by rehashing the time savings discussion, but not really anything new. Perhaps the literature tells us that all public transit is slow and then you find we have a good alternative in vans? I’m sorry to repeat myself, but context would help the authors understand if this is an important finding or not. 

Line 310. This is interesting and warrants discussion here because it raises a lot of interesting questions. Is this extra service illegal? Is it that common? What does this matter? I guess the service areas are not so constrained. 

 Line 326 Here you say vans complement service. Before you said compete. 

 Line 336. You raise this first/last mile point earlier, but you never talk about connections and their importance to the vans. What percentage of people do transfer?

Conclusion

The point about the integration of public and private is  potentially a good one, but it is not really discussed in the paper. Is it just the buses are too full and thus slower and so these are just private bus lines?

Author Response

Please find attached a Word document that provides an overall summary of our revisions followed by point-by-point responses to all reviewer comments (including yours).

Round  2

Reviewer 2 Report

Dear authors,

Thank you for modifing the articule. Now the paper is much well presented. 

I have no further comment on it. 

Author Response

Thank you for your good suggestions in the first round of review - our paper is much better because of them!

Reviewer 3 Report

This paper is much improved and needs only a few clean up revisions. The added context makes this a much more interesting paper.

1. In tables, use standard asterick (*) system to indicate significance. The bolding and coloring may not work out well if I decide, for example, to print out the pages on my black and white printer. 

2. Table 1: The comparison with Brooklyn is not well explained. Why do I care? I think that you confirm some of that survey’s findings later. But do we need to see the results. I am not so sure. You can keep it, but then use it more in the paper.

3. I would be interested in seeing a comparison to the neighborhood demographics, which are somewhat described in the paragraph starting on page 7 line 189. This helps me gauge whether the van riders reflect the neighborhood. You could discuss that for a paragraph. 

4. Table 3: (1) You need to explain that you actually ran four chi-square tests, one for each of the demographic variables on the y-axis. Just a line in the text so that no one things you ran this entire table at once. (2) You cannot run the y-axis variables (transportation choices) in a chi-square because one of the assumptions of that test is that the values of transportation choices are mutually exclusive. These values are not because a respondent could have a car available and have a metro card and use van more than once a week, etc. You could do a series of 2X2 tests – income v car yes/no and then income v metro card yes/no, etc. (And if you did this already, then just explain it better so someone does not presume you did the analysis incorrectly.)

You have the same issue with Tables 5 & 6. I love the analysis that you bring forward. You either have to explain it better or sharpen the significance testing. In all of these, if needed, 2X2 tests would be fine along with a description of what you did.

5. Page 11 line 282 –How do we know bus rider habits? You mention later that you road bus routes once. This is an important limitation. While I find the difference interesting, I’m not sure it is worth mentioning with just one ride. If you like it enough, then you just need to re-order how you present it so we know you road the bus and add a larger caveat. Perhaps, if important as “something to be explored.” If this data actually comes from MTA records, then you need to tell me. 

6. Page 12 line 317 This paragraph is good. Very easy to understand and puts the cost finding into context. (I didn’t find it “remarkable,” but the analysis makes sense and the authors can categorize as they want.)

7. Page 12, last paragraph. This discussion seems to duplicate the discussion started on line 317, where I think it is done better. 

8. Nice revision of section 5.4. Much clearer and helps me understand the situation. 

9. Section 6.1 is a nice conclusion section. You might merge it with section 7. 

10. In the appendix, you might condense the raw data to one page. No real need to see the actual instrument. Just questions and answer tallies. 

Author Response

This paper is much improved and needs only a few clean up revisions. The added context makes this a much more interesting paper.

Thank you. We really appreciated your suggestions in the first round of revisions, and agree that the paper is much improved!

1. In tables, use standard asterick (*) system to indicate significance. The bolding and coloring may not work out well if I decide, for example, to print out the pages on my black and white printer.

We have revised the tables to use both the asterisk system (as suggested here), and to include the chi2 test statistic and number of observations for each test. This should make it clear exactly what tests were done and the outcomes. We have retained the boldface and shading as well, because – especially now that there are so many more numbers in the tables – we think this makes it much easier to focus in on the percentages that are significantly distinct from each other.

2. Table 1: The comparison with Brooklyn is not well explained. Why do I care? I think that you confirm some of that survey’s findings later. But do we need to see the results. I am not so sure. You can keep it, but then use it more in the paper.

We see your point here, but we would like to keep the Brooklyn study summary statistics in our paper. Part of the reason for this is that it took more effort than it should have to extract them from the PhD dissertation they came from. The dissertation does not include summary statistics from the survey in table format, so putting these data together required carefully reading quite a bit of text to find the relevant information. Including the Brooklyn survey information here allows other researchers to more easily access it.

In addition, the Brooklyn commuter van study was extremely similar to our own, and there aren’t any others that we know of. On many metrics, the two survey samples have similar characteristics, and clearly illustrating this is useful, we think.

3. I would be interested in seeing a comparison to the neighborhood demographics, which are somewhat described in the paragraph starting on page 7 line 189. This helps me gauge whether the van riders reflect the neighborhood. You could discuss that for a paragraph.

We have added the following sentence to that paragraph to quantify the comparison with neighborhood demographics.

“Data from the American Community Survey (2016) show that approximately 55% of the population in the study area were female, 50% were immigrants who commuted to work, 43% commute to work via transit, 60% of the households reported incomes over $50,000, and 81% of households owned at least one car.”

4. Table 3: (1) You need to explain that you actually ran four chi-square tests, one for each of the demographic variables on the y-axis. Just a line in the text so that no one things you ran this entire table at once. (2) You cannot run the y-axis variables (transportation choices) in a chi-square because one of the assumptions of that test is that the values of transportation choices are mutually exclusive. These values are not because a respondent could have a car available and have a metro card and use van more than once a week, etc. You could do a series of 2X2 tests – income v car yes/no and then income v metro card yes/no, etc. (And if you did this already, then just explain it better so someone does not presume you did the analysis incorrectly.)

Thanks for asking this question. In response to this and your first comment (above), we have added the chi2 test statistic, asterisks to indicate statistical significance level, and the number of observations for each test that we calculated. All of the tests are of the “2X2” variety, which should now be crystal clear from the way the table is displayed. We have also revised the text to be clearer about our statistical tests.

You have the same issue with Tables 5 & 6. I love the analysis that you bring forward. You either have to explain it better or sharpen the significance testing. In all of these, if needed, 2X2 tests would be fine along with a description of what you did.

We have corrected Tables 5 and 6 in the same way.

5. Page 11 line 282 –How do we know bus rider habits? You mention later that you road bus routes once. This is an important limitation. While I find the difference interesting, I’m not sure it is worth mentioning with just one ride. If you like it enough, then you just need to re-order how you present it so we know you road the bus and add a larger caveat. Perhaps, if important as “something to be explored.” If this data actually comes from MTA records, then you need to tell me.

We have taken this mention of “bus rider habits” out of the text. We agree that we don’t have enough bus rider data to make this comparative statement. We retain the point about van riders often riding the full length of the route (which was the important part, anyway).

6. Page 12 line 317 This paragraph is good. Very easy to understand and puts the cost finding into context. (I didn’t find it “remarkable,” but the analysis makes sense and the authors can categorize as they want.)

Thank you.

7. Page 12, last paragraph. This discussion seems to duplicate the discussion started on line 317, where I think it is done better.

We are not certain what you are referring to here. The last full paragraph on page 12 focuses on the revenue impact of the commuter vans on the MTA, which is quite different from the point about van riders being willing to pay a premium for quality service – in spite of the fact that many of them are quite poor. The last paragraph on page 12 summarizes our conclusion that “the Eastern Queens commuter vans complement MTA transit service in two important ways”.

We spent time searching for places where we repeated the point about low-income van riders’ willingness to pay for quality service so make sure we weren’t being too redundant. We only found this repeated in the beginning of the discussion section, however, where we are repeating our main findings in a condensed way (so that repetition was on purpose).

8. Nice revision of section 5.4. Much clearer and helps me understand the situation.

Thank you.

9. Section 6.1 is a nice conclusion section. You might merge it with section 7.

We see your point here, but we’re not going to take this suggestion, and here’s why. We think that the subheading for section 6.1 is important to include, and if we merged it with section 7 (making 6.1 + 7 into our “conclusion” section), then we’d lose that descriptive subheading. Since they follow each other now, it is nearly the same effect as if they were one section.

10. In the appendix, you might condense the raw data to one page. No real need to see the actual instrument. Just questions and answer tallies.

We’d like to keep the full instrument in the appendix. That way, if other researchers would like to replicate our work, they can use it as a starting point.